# Generation of Peptides for Highly Efficient Proximity Utilizing Site-Specific Biotinylation in Cells

**DOI:** 10.3390/life12020300

**Published:** 2022-02-16

**Authors:** Arman Kulyyassov, Yerlan Ramankulov, Vasily Ogryzko

**Affiliations:** 1Republican State Enterprise “National Center for Biotechnology” under the Science Committee of Ministry of Education and Science of the Republic of Kazakhstan, 13/5 Kurgalzhynskoye Road, Nur-Sultan 010000, Kazakhstan; ramanculov@biocenter.kz; 2UMR8126, Institut de Cancerologie Gustave Roussy, Universite Paris-Sud 11, CNRS, 94805 Villejuif, France; vasily.ogryzko@gustaveroussy.fr

**Keywords:** biotin acceptor peptide (BAP), protein–protein interactions (PPI), proximity utilizing biotinylation (PUB), site-specific biotinylation

## Abstract

Protein tags are peptide sequences genetically embedded into a recombinant protein for various purposes, such as affinity purification, Western blotting, and immunofluorescence. Another recent application of peptide tags is in vivo labeling and analysis of protein–protein interactions (PPI) by proteomics methods. One of the common workflows involves site-specific in vivo biotinylation of an AviTag-fused protein in the presence of the biotin ligase BirA. However, due to the rapid kinetics of labeling, this tag is not ideal for analysis of PPI. Here we describe the rationale, design, and protocol for the new biotin acceptor peptides BAP1070 and BAP1108 using modular assembling of biotin acceptor fragments, DNA sequencing, transient expression of proteins in cells, and Western blotting methods. These tags were used in the Proximity Utilizing Biotinylation (PUB) method, which is based on coexpression of BAP-X and BirA-Y in mammalian cells, where X or Y are candidate interacting proteins of interest. By changing the sequence of these peptides, a low level of background biotinylation is achieved, which occurs due to random collisions of proteins in cells. Over 100 plasmid constructs, containing genes of transcription factors, histones, gene repressors, and other nuclear proteins were obtained during implementation of projects related to this method.

## 1. Introduction

Proteins are the main macromolecules that play a key role in the life of the cell. Therefore, the ability to observe protein function in vivo is a key research area in modern biology. Among the most promising ways to achieve this goal is the use of site-specific labeling [1]. This method uses various enzymes, such as various peptidases [2,3], transferases [4], or ligases [5,6], which recognize specific amino acid sequences and label them with a covalent tag. One of the most commonly used enzymes is the biotin ligase BirA, which is used to label the AviTag target [7,8]. The choice of this method is based on the possibility to achieve a high degree of purification of biotinylated proteins and a wide range of commercially available reagents and kits.

Recently, site-specific labeling methods have been used in a wide range of practical applications, such as improving the technology for purification of recombinant proteins [9], creating multifunctional protein complexes [10,11], and in clinically important conjugates with polymers and fluorescent molecules [12,13]. Some enzymes used for these purposes in vitro can be engineered to modify reaction kinetics to increase the speed and efficiency of substrate labeling [10]. However, when studying protein–protein interactions (PPI) in the context of a living cell, it is necessary to use site-specific labeling of proteins with slower and linear labeling kinetics. This is due to the fact that in a living cell, all expressed proteins have a wide range of concentrations [14,15,16] and are rather highly crowded and dynamic [17]. Therefore, when studying PPIs, a high level of background labeling is very often observed due to random collisions of molecules. To address this problem, we designed the new biotin acceptors BAP1070 and BAP1108, which have a lower substrate specificity compared to the AviTag peptide. These peptides can be used in Proximity Utilizing Biotinylation (PUB) [18]. The principle of the PUB method is based on using the enzyme/substrate pair reaction [7,19,20], where two proteins to be tested for their interaction in vivo are coexpressed in mammalian cells, with one as fused to the BAP and the other fused to an *Escherichia coli* biotin ligase (BirA) [8]. When the two proteins are in proximity to each other (such as when an interaction occurs), more efficient biotinylation of the BAP is expected. The biotinylation status of the BAP fusion protein can be further monitored by Western blot, mass spectrometry, or confocal microscopy.

The aim of this study was design and generation of efficient biotin acceptor peptides for study of protein–protein interactions in cells.

## 2. Materials and Methods

### 2.1. Materials

DNA Ladder for electrophoresis (Fermentas “GeneRules DNA ladder mix”, 100–10,000 base range, or similar, #SM1173), XbaI restriction enzyme (New England Biolabs, Ipswich, MA, USA, #R0145), XhoI restriction enzyme (New England Biolabs, #R0146), NotI restriction enzyme (New England Biolabs, #R0189), BamHI restriction enzyme (New England Biolabs, #R0136), BglII restriction enzyme (New England Biolabs, #R0144), Deoxynucleotide (dNTP) solution mix (Thermo Fisher, Rockford, IL, USA, #R0192), T4 DNA ligase (Thermo Fisher, #15224025), Glycerol (Sigma-Aldrich, St. Louis, MO, USA, #G5516), Agarose for gel electrophoresis (Lonza, Basel, Switzerland, #50004), TAE buffer (50X, Applichem, #A4686,1000), Montage gel extraction kit (Millipore, St. Louis, MO, USA, #LSKGEL050), Pwo DNA polymerase (Roche, Welwyn Garden City, UK, 11644947001), Millipore PCR purification kit (#P36461, Rev. A, 03/05), Ampicillin (Sigma-Aldrich, MO, USA #A9518), Agar (BD Difco, Davenport, IL, USA, #281210), Tryptone (BD Bacto, MO, USA, #211699), Yeast extract (BD Bacto, #212720), QIAprep Spin Miniprep Kit (QIAGEN, #27106), GenElute™ Endotoxin-free Plasmid Midiprep Kit or similar (Sigma, MO, USA, #PLED35-1KT), QIAGEN Plasmid Maxi Kit (QIAGEN, MD, USA, #12362). The *E. coli* DH5α competent cells were prepared according to the Hanahan Method [21].

Preparation of incubation buffer (10×): 2.5 mL of 2M Tris-HCl, pH 7.5 and 1 mL of 1 M MgCl_2_ solution were loaded into a 15-mL Falcon tube. Then, UltraPureDNAse and RNAse-free water were added to a final volume of 10 mL and mixed.

Acrylamide Gel Elution Buffer was prepared from stock solutions for a final concentration of 500 mM ammonium acetate, 0.1% SDS, 1 mM EDTA, and 10 mM magnesium acetate at pH 8.0.

Preparation of stock solutions of ATP and DTT required for ligation: 8 μL of a 0.5 M DTT solution was mixed with 92 μL of water to obtain 100 μL of a 40 mM DTT solution. A total of 40 μL of a 100 mM ATP solution was mixed with 60 μL of water to obtain 100 μL of a 40 mM ATP solution.

Preparation of 10 mL non-denaturing 15% Polyacrylamide Gel (PAAG): 3.75 mL of 40% (29:1) acrylic/bisacrylamide, 1.0 mL of 10×x TBE buffer, and 5.25 mL of H_2_O were mixed and stirred in a 15-mL Falcon tube. The polymerization initiators TEMED (12 μL) and 10% APS (ammonium persulfate, 120 μL) were then added and mixed. Then, using a pipette, the mixture was loaded into the electrophoresis cassette; a comb was inserted to form the wells and was left for 30 min. After gel polymerization, the comb was removed, and the wells were carefully washed first with distilled water and then with TBE buffer.

HEK293T (Human embryonic kidney), MRC-5 fetal lung fibroblasts, or HeLa cell lines from American Type Culture Collection (ATCC) can be used for the experiments. Dulbecco’s Modified Eagle Medium (DMEM) containing 1% penicillin-streptomycin, 10% FBS, and 1% L-glutamine was used for growing cells. FuGENE HD Transfection Reagent (Promega, Madison, WI, USA, #E2311) was used for transfection. The stock solution for biotin labeling (1 mg/mL) was prepared by adding 0.5 mL of 410 mM NaOH to 50 mL of suspension containing 50 mg of biotin (Sigma-Aldrich, #B4501-1G) in water.

### 2.2. Workplan A

To prepare the BAD 1070 fragment, four oligonucleotides marked as His, Hisas, BAD1070 forward, and BAD1070reverse and a primer BADPCR were ordered from Operon Company (Table 1). The primer BADPCR was used further to select the obtained clones by colony PCR. Tubes were spun and primers dissolved in water to a final concentration of 100 μM.

#### 2.2.1. Vector Preparation

The tube containing 1 μg (0.4 μL) of pOzFHHN-H2Az vector, 2 μL of 10×xNEBuffer 3.1 and 15.6 μL of water was heated 5 min at 65 °C and cooled on ice. Then, 1 μL of BglII and 1 μL of XhoI enzymes were added to the solution. The tube was vortexed, centrifuged, and incubated at 37 °C for 3 h. Enzymes were deactivated by heating 10 min at 65 °C. The digested products were loaded on 0.7% agarose and electrophoresis was run in TAE buffer (55 V, 90 min) in a Mini-Sub Cell GT Horizontal Electrophoresis Apparatus (Bio-Rad, #1640300). After 5 min treatment of the gel with ethidium bromide (0.5 μg/mL), DNA bands corresponding to the size of the pOzFHHN.H2Az vector were cut under UV light using an ECX-15.M UV Transilluminator. A Millipore gel extraction kit for DNA extraction was used to isolate restriction products. The gel and column tubes were centrifuged at 5000× *g* at 4 °C for 10 min. The resulting DNA solution (22 μL) was heated at 65 °C for 15 min.

#### 2.2.2. Construction of BAP1070 DNA Fragment

Before ligation with a vector, a fragment was obtained from oligonucleotides using the following steps in the following order: annealing, ligation, electrophoresis, separation, elution, dilution, and ligation with vector (Figure 1).

Preparation of equimolar amounts of oligonucleotides BAD1070: 2 μL of 100 μM solutions of primers His, HisAs, BAD1070forward, BAD1070reverse, 2.2 μL of 10×x incubation buffer, and 11.8 μL of deionized water was mixed in a 0.5-mL Eppendorf tube (Total volume 22 μL).

Annealing: The tube was heated at 95 °C for 5 min in a PCR machine and placed in water preheated to 95 °C. The water bath was turned off to allow for slow annealing of oligonucleotides at lower temperatures from 95 °C to 45 °C (approximately 1.5 h).

Ligation of oligonucleotides: 0.5 μL of ATP and 0.5 μL of DTT were added to a tube with 20 μL of a mixture of oligonucleotides, heated to 65 °C for 30 s, and cooled on ice. Then, 1 μL of T4 Invitrogen DNA ligase was added to the tube, vortexed, and microcentrifuged. The sample tube was left overnight at 14 °C.

Electrophoresis separation: The gel cassette was placed in an electrophoresis chamber and the contents of tube with the mixture after ligation of the oligonucleotides were loaded into the gel. Promega 10 bp DNA Ladder was used as a marker. Electrophoresis was performed in TBE buffer solution at 50 volts for 5 h. After electrophoresis, the gel was developed with a solution of ethidium bromide for 5 min and washed with water. The bands corresponding to a mass of 84 kb were then excised in UV light. Pieces of the gel were crushed using a blade and transferred to a 1.5-mL Eppendorf tube.

Elution: The crushed pieces of gel were incubated with shaking on a shaker overnight with Acrylamide Gel Elution Buffer at 37 °C [22]. The next day, the supernatant was removed using a pipette and transferred to a clean tube and used for ligation with the vector.

Dilution: 2 μL of selected aliquots of oligonucleotide mixtures from the tube were diluted with water at10^−1^, 10^−2^, 10^−3^, and 10^−4^ for subsequent ligation with the vector.

Ligation with vector: Three clean 0.5-mL Eppendorf tubes (marked # 1, #2, and #3) were loaded with 2 μL of the 10^−2^, 10^−3^, and 10^−4^ dilution of the oligonucleotides. A master mix of the following composition was then prepared: 4 μL of vector, 4 μL of 10×x incubation buffer, 1 μL of ATP (1:40), 1 μL of DTT (1:40), and 20μL of water (total volume 30 μL). The mixture was vortexed and spun. A total of 7 μL solution was transferred to each of the three tubes containing 2 μL of oligonucleotides. All tubes were heated to 65 °C for 30 s and cooled on ice. In each tube, 1 μL of T4 DNA ligase was added and left overnight at 14 °C. Colony PCR for selection of correct clones with plasmid pOzFHHN-BAP1070-H2Az after transformation was performed according to Green protocol [23].

#### 2.2.3. Subcloning of Fragment, Containing BAP1070-H2Az into pcDNA3.1(+) Vector

The tube containing 1 μg of commercial pcDNA3.1(+) vector (Invitrogen, #V79020), 2 μL of 10×xNEBuffer 3.1, and 15 μL of water was heated at 65 °C for 5 min and cooled on ice. Then 1 μL of BamHI and 1 μL of XbaI enzymes were added to the solution. The tube was then vortexed, centrifuged, and incubated at 37°C for 3 h.

The tube containing 1 μg (0.5 μL) of pOzFHHN-BAP1070-H2Az vector, 2 μL of 10×xNEBuffer 3.1, and 15.5 μL of water was heated at 65 °C for 5 min and cooled on ice. Then 1 μL of BglII and 1μL of XbaI enzymes were added to the solution. The tube was then vortexed, centrifuged, and incubated at 37 °C for 3 h. The protocol is identical to that described previously in Section 2.2.1.

Ligation of vector with fragment and bacterial transformation were performed as described in Section 2.2.2. Plasmid midi- (or maxi-) prep DNA preparation was performed according to the manufacturer’s instructions.

### 2.3. Workplan B

To obtain a plasmid with a new variant of the biotin acceptor peptide BAP1108, we designed primers containing recognition sites for restriction endonucleases KpnI and XbaI, which allowed cloning at these sites into the vector plasmid pcDNA3.1 (+).

Primers 1 and 2 were ordered from Operon company (Table 1). Plasmid pOz-BAP1070-H2Az was used as a template for the first PCR amplification of BAP1108-H2Az. PCR conditions were the following: denaturation at 94 °C for 15 s, annealing at 54 °C for 30 s, elongation at 72 °C for 1 min, 20 cycles. Primers 2 and 3 were used for the first PCR.

After the first PCR, the reaction products were separated by electrophoresis in a 0.8% agarose gel and stained with ethidium bromide. A fragment corresponding in size to the amplicon (approximately 400 bp) was excised with a scalpel. The excised piece of gel containing the amplicon was then placed into tubes from the Millipore agarose gel DNA isolation kit and centrifuged at 5000× *g* for 10 min. The resulting solution with the purified amplicon was used for the second PCR.

Primers 1 and 3 were used for the second PCR. Preparation of master mix and PCR conditions were optimized according to instructions from the Pwo polymerase manufacturer. The PCR reaction products (10 μL, or 1/5 of the volume) for visualization were subjected to electrophoresis in 0.8% agarose gel and stained with ethidium bromide. The remaining PCR products (40 μL) were placed in tubes with filter elements from the Millipore PCR purification kit and 300 μL of water was added. The tube was centrifuged at 1000× *g* for 15 min to remove salts and primers. Then, 20 μL of water was added to the filter element containing the amplified DNA, inverted, inserted into a new tube from the set, and centrifuged at 1000× *g* for 2 min.

The resulting amplicon BAP1108-H2Az was cloned into the pcDNA3.1(+) vector. Vector DNA pcDNA3.1(+)-BAP1070-HP1γ (1 μg) and the PCR amplification product BAP1108-H2Az were incubated with restriction enzymes KpnI and XbaI (20 u.a. in 20 μL of the reaction mixture) in the corresponding buffer solutions at 37 °C for 2 h. The digested products were separated by electrophoresis in 0.8% agarose gel and stained with ethidium bromide. Fragments visualized in UV light were excised and DNA was extracted from the agarose gel as described in Section 2.2.1. Ligation of the vector and DNA fragment obtained from the previous stage was performed using T4 DNA ligase overnight at 14 °C.

After transformation of *E. coli* competent cells (strain DH5α) and selection of clones carrying recombinant plasmids with the inserted gene, the plasmid pcDNA3.1(+)-BAP1108-H2Az was obtained. H2Az can be replaced by other genes (GFP, HP1α, Sox2, etc) using restriction enzymes XhoI and NotI.

### 2.4. Cell Culture, Transient Transfection, and Biotin Labeling In Vivo (Generalized Protocol)

HEK293T cells were cultured in DMEM according to ATCC protocols and dispersed in 6-well plates at 2 × 10^5^ cells/well 24 h prior to transfection. Cells at approximately 70% to 80% confluency were used before transfection. At least 1 h before transfection, the growth medium was replaced with 1.5 mL of fresh DMEM medium. For transient transfection, a DNA solution free of protein, RNA, chemical, and endotoxin contamination with an A260/A280 ratio of 1.8 (or higher) was used. An important point in the protocol is the use of relative amounts of plasmids for transfection. The BAP-X target proteins should be expressed at higher levels in the cell in comparison to BirA-Y fusions. For this purpose, we have used pcDNA3-BAP-X (expression under the strong CMV promoter) and pOz-BirA-Y (expression under the weak MoMuLV promoter). Typically, the following amounts of plasmids were used for transfection: 0.5 μg for pcDNA3-BAP-X and 0.3 μg for pOz-BirA-Y. The BirA-fused vector contains humanized wild-type version of *Escherichia coli* biotin ligase [24].

Plasmids were diluted in sterile deionized water at up to 100 μL for each tube. A FuGENE vial was mixed well by inverting and 8 μL of FuGENE HD Transfection Reagent was added directly into the solution containing plasmids without allowing contact with the walls of the plastic tubes. Then solutions were mixed carefully by pipetting (10–15 times) and incubated at room temperature for 40 min. After formation of the transfection complex, the solution was added dropwise to the cells and incubated in a CO_2_ incubator at 37 °C for 48 h.

For the biotin labeling in vivo (5 min, 30 min, 3 h, 24 h), a stock solution of biotin (1 mg/mL) was added to a final concentration of 5 μg/mL. The pH was stabilized by addition of 50 mM HEPES (pH 7.35) to the medium.

Along with commercial FuGENE or Lipofectamine, the classical calcium phosphate method may also be applied for expression of target and BirA fusions of the protein of interest. Usually, in the PUB method the cells are harvested 40 to 48 h after transfection, which is optimal for detection and monitoring of protein–protein interactions.

### 2.5. Cell Lysis and Sample Preparation

0.5% Triton X-100 in Cytoskeleton (CSK) buffer was used to disrupt cells and isolate nuclei as previously described [18]. The degree of disruption and purity of the nuclear fractions was monitored using light microscopy. The Western blotting protocol is described in our Appendix A.

## 3. Results

### 3.1. Assembling of BAP1070 Fragment (Workplan A)

The assembling consists of six steps, including annealing, ligation, electrophoresis, elution, dilution, and ligation with vector (Figure 1A). In the first step, we used annealing with a slow decrease in temperature from 95 °C to 45 °C, which can provide maximum efficiency for pairwise annealing of the four DNA oligonucleotides (His, Hisas, BAD1070 forward, and BAD1070 reverse fragments). The incubation buffer for oligos was ATP free due to instability of ATP during prolonged incubation at high temperatures [25]. The ATP was added before adding DNA ligase. For ligation, two phosphorylated fragments and two non-phosphorylated fragments were used. This combination of oligonucleotides prevented the possibility of adverse reactions that could significantly reduce the yield of the target product. The ligation products were then separated by polyacrylamide gel electrophoresis and the bands containing the desired oligonucleotides corresponding to the expected masses were excised from the gel. The DNA was eluted from the gel according to a standard protocol [22]. This strategy, with two steps, favors modularity and can be used to construct new fragments reusing two of the oligos.

Various aqueous dilutions (10^−1^, 10^−2^, 10^−3^, and 10^−4^) of the BAP1070 fragment with the sticky ends BglII and XhoI were ligated with the precut pOzFHHN-H2Az vector overnight at 14 °C. The ligation products were used for bacterial transformation of *Escherichia coli* DH5α competent cells. The resulting clones containing recombinant DNA were selected by colony PCR, and the selected strains were used for the preparation of minispin plasmid DNA and subsequent sequencing. The optimal dilution rates of BAP1070 fragments were 10^−1^, 10^−2^, and 10^−3^. After selection, sequence confirmation, and midiprep preparation the target plasmid contained BAP1070 fused to downstream H2Az under the weak MoMuLV promoter. We also subcloned fragment BAP1070-H2Az into another commercial vector pcDNA3.1(+), which contains the strong CMV promoter.

### 3.2. Assembling of BAP1108 Fragment (Workplan B)

Two sequential PCR steps were used to generate the DNA sequence of this peptide (Figure 1B). In the first PCR step, pOzFHHN-BAP1070-H2Az obtained in the previous experiments was used as a template. In the second PCR, the amplicon from the first PCR was used as a template to obtain the full-length BAP1108-H2Az fragment. This fragment was subcloned into pcDNA.3.1(+) vector with a strong promoter. The entire inserts in the final vectors were sequenced. Genes of any protein of interest (ORF Y) can be cloned into these vectors between the XhoI and NotI sites (Figure 1A,B).

## 4. Discussion

### 4.1. Site-Specific Biotinylation in the Model System BirA + BAP-X

Wild-type biotin ligase BirA is very convenient for site-specific biotinylation in vivo because this modification is relatively rare among proteins found in various organisms [26,27]. In earlier works, pairs of recombinant BirA and BAP-POI (or AP-X) proteins were mainly expressed in cells, where BAP peptide is the biotin acceptor peptide tag and POI is the protein of interest [7,28]. The main goal was quantitative in vivo labeling of tagged proteins, followed by affinity purification from cell lysates using commercially available streptavidin beads. Due to the strong binding of biotin with streptavidin, and using buffer solutions containing detergents and chaotropic agents, a high degree of purification of labeled proteins could be achieved. This can be particularly useful for studying protein-DNA interactions using ChIP and obtaining results with a better signal-to-noise ratio [29,30]. Using less stringent conditions and buffer solutions similar to physiological conditions allows the isolation of bound proteins and protein complexes for subsequent identification by mass spectrometry, as in the affinity (or tandem affinity) purification AP (or TAP)-MS method. For example, proteins associated with the Nanog pluripotency factor [31], or the RNA binding protein RALY [32] were identified using coexpression of these AP-tagged proteins with BirA. However, this approach did not give significant advantages over the traditionally used methods of protein complex analysis [33,34].

### 4.2. Site-Specific Biotinylation in the Model System BirA-X + BAP-Y

A more interesting variant of the experiments is the expression of two fusion proteins BirA-X + BAP-Y. In this case, biotinylation is the direct result of the interaction or proximity of the X and Y proteins. Fernandez et al. [19] demonstrated the suitability of this approach using the rapamycin-regulated interaction of FKBP (FK506 binding protein) and FRB (FKBP-rapamycin binding protein). Coexpression of FKBP-AP(-3) and FRB-BirA resulted in a 5-fold increase in biotinylation in the presence of rapamycin compared to controls where rapamycin was not added. The AP(-3) is a acceptor peptide designed with replacement of three amino acid residues at the C-terminus of the Avitag peptide, which led to a decrease in the background level of biotinylation.

Another variant of this approach is PUB method [18] with some main features method including:Reduced substrate specificity of biotin acceptor domain in BAP;Addition of His-tag to the sequence of BAP;Addition of two flanking arginines for generation of tryptic peptides and monitoring of biotinylation level of BAP by LC-MS/MS;Using humanized wild-type biotin ligase BirA;Coexpression in mammalian cells from 2 plasmids instead of a single bicistronic vector;Availability of vectors with strong (CMV) and weak (MoMuLV) promoters;Plasmid constructs have elements for both transient and stable expression in mammalian cells.

Reduced substrate specificity of BAP allowed decreasing background biotinylation and enhancing difference between experiment and control in comparison with Avitag. Addition of His-tag helps to purify the BAP-Y proteins from cell lysates regardless of their biotinylation status and evaluate the total amount either by Western blotting or MRM mode of LC-MS/MS. Wild-type BirA from *Escherichia coli* was codon-optimized for using in transfection in mammalian cell lines [24]. Using transfection with two plasmids was more appropriate and similar to physiological conditions contrary to bicistronic design which causes high local concentration of BirA-X fusions and BAP-Y targets in cell. For correct analysis and interpretation of results the concentraion of enzyme should be much less than concentration of target [BirA-X] << [BAP-Y]. This difference can be achieved by using vectors with different promoters. For example BirA-X is expressed by plasmids with weak MoMuLV promoter and BAP-Y is expressed by CMV promoter. Plasmids used for PUB contained also EMC-IRES and IL2R alpha elements to have option to generate cell lines for stable expression of recombinant proteins.

### 4.3. Design, Generation, and Properties of BAP Peptides

To use biotin acceptor peptides to study protein–protein interactions in cells, we required a peptide with more linear biotinylation kinetics. This could be achieved by reducing the substrate specificity of this peptide by redesigning its amino acid sequence. Based on literature data [35,36] and the results of screening of large peptide libraries for activity in BirA-catalyzed biotinylation and identification of a “consensus peptide”, we proposed the new sequences BAP1070 and BAP1108 that may be expected to have less reaction ability (Figure 2).

We took the AviTag peptide and changed its sequence as follows. The three amino acids at positions +3, 4, and 5 were replaced by RGG, since removal of WHE significantly reduces background biotinylation as demonstrated by Fernandez-Suarez et al. [19]. To further reduce substrate specificity, we also made amino acid substitutions at position −4 (F to L) and at position +2 (E to V or H). To facilitate purification from the cell lysate or to monitor the entire amount of target BAP peptides, we also introduced a 7×His tag at position −16.

To reduce the cost of synthesis, instead of ordering one long 87-mer oligonucleotide, we ordered four short oligonucleotides, including two regular (His and BAD1070 reverse) and two phosphorylated forms (Hisas and BAD1070 forward). After assembly of the whole BAP1070 oligonucleotide, we subcloned it into the pOz and pcDNA3.1 vectors to have the option to express proteins under weak and strong promoters, respectively. This may be important to express recombinant proteins at concentrations close to physiological levels of endogenous proteins in cells. Both types of vectors contain a bicistronic transcriptional unit that allows for expression of two proteins from a single transcript, specifically one from the gene of interest and also the α chain of a cell-surface antigen, interleukin-2 receptor (IL2Rα). This vector was designed for the creation of stable expression cell lines [37].

A model experiment similar to the one described earlier [18] was used to evaluate the substrate specificity of the new peptide BAP1108 (Appendix A). BirA-GFP and BAP-GFP were coexpressed in HEK293T cells to estimate the level of random collisions between noninteracting GFP proteins in cells that will result in background biotinylation. Both BAP1070 and BAP1108 peptides showed similar linear dynamics of biotinylation in the HEK293T cells. The time of half-saturation was 45 (BAP1070) and 48 times (BAP1108) greater in comparison with parallel cotransfections of BirA-GFP and AviTag-GFP (680 and 720 min vs. 15 min). Rapid saturation in the BAP biotinylation levels can mask dissimilarities in biotinylation efficiency, therefore the slower kinetics of BAP biotinylation was preferred.

Since the PUB is LC-MS/MS oriented method, experiments were also performed to identify and analyze the new peptide BAP1108 using mass spectrometry. As previously shown [38], the use of on-bead digestion in sample preparation containing BAP1070 reduced the time and number of steps in the protocol. For example, the steps of imidazole elution, polyacrylamide gel electrophoresis, staining and destaining, gel cutting, and subsequent washing of gel pieces were eliminated. On-bead digestion produced fairly intense peaks of propionylated and biotinylated forms of BAP1070 peptide on MRM chromatograms. Treatment by propionic anhydride was used to modify target lysine [39] in non-biotinylated BAP peptide and prevent from proteolysis by trypsin. This resulted in peptides with comparable sizes and facilitated the analysis of MRM data. However, in the case of peptide BAP1108, it was not possible to detect peaks and MS/MS spectra corresponding to this peptide. Since BAP1108 was obtained by replacing the valine in BAP1070 with histidine, this BAP1108 peptide remained bound on the surface of Ni-sepharose beads during on-bead digest conditions. Therefore, in this case, we used the standard protocol including imidazole elution and electrophoresis as described earlier [18]. We detected and identified both propionylated and biotinylated forms of new BAP1108 peptide from experiments where BirA-GFP and BAP-GFP were coexpressed in HEK293T cells using MRM method (Appendix A).

### 4.4. Examples of Applications of PUB Method

We constructed and obtained many plasmids containing fusions of BAP with various nuclear proteins and tested in different PPI model systems including protein oligomerization (Tap54α vs. HP1γ) or ordinary protein–protein interaction (Kap1 and HP1). The advantage of linear biotinylation kinetics of BAP was demonstrated also for models of protein–protein proximity [18] which can be a result of colocalization in the nucleus (“PCNA+H3.1” vs. “PCNA+CenpA”), different subnuclear domains (macroH2A vs. H2ABBD) or intracellular compartmentalization (GFP and HP1).

These encouraging results helped to modify the PUB method further to an approach termed as PUB-NChIP (proximity utilizing biotinylation with native ChIP) to purify and study the protein composition of chromatin in proximity to a nuclear protein of interest (Figure 3A), for example, Rad18 as model protein [40].

An interesting application of the PUB method was demonstrated in a paper published recently [41]. Authors of this paper described a novel, low-cost, reliable, sensitive, and practical microscopy-based single-cell approach, termed Topokaryotyping. Using this method they reconstructed and analyzed the interphase positioning of genomic loci relative to a given nuclear landmark (Figure 3B).

This BAP/BirA pair reaction was also successfully applied by another research group in a recent paper published in Cell (Figure 3C) where an inducible, proximity-dependent labeling system was developed to label nucleosomes that contained replication-dependent H3.1 and H3.2 histones at desired loci in embryonic stem cells [42].

We recently demonstrated the application of the PUB method for the detection of in vivo transient DNA-dependent protein–protein proximity of pluripotency transcription factors, Sox2 and Oct4 [38]. Although Sox2 is essential during mammalian embryogenesis it is also known factor playing a dramatic role in tumor growth and drug resistance [43]. Sox2 has HMG domain, and Oct4 has two POU_HD_ and POU_S_ domains that help these transcription factors to assemble on composite DNA motifs of regulatory elements of genes such as *Utf1* or *FGF4* [44]. Due to this DNA-dependent binding and interaction of these proteins on such motifs, this results in the labeling of the BAP target and high level of biotinylation detected by MRM mode [45] of LC-MS/MS and streptavidin Western blotting in comparison with controls, where we used GFP or Tap54beta fusions of BAP (Figure 3D). Quantification of MRM data was performed using Skyline software [46]. It is interesting to note that, the using of another widely applied method, such as Co-immunoprecipitation (Co-IP) in the search of Oct4 interactors did not reveal one of the best-studied partners of Oct4, namely Sox2 [31]. This is probably explained by the fact that Oct4 and Sox2 interact most stably when bound to neighboring sites on DNA.

### 4.5. Differences in Two Approaches and Perspectives of BirA-Catalyzed In Vivo Labeling for Biological Research

One major difference should be noted between the two approaches described in Section 4.1 and Section 4.2. This difference lies in the choice of substrate or target for BirA. In the former, Avitag peptide is used, with fast biotinylation kinetics, resulting in the quantitative yield of the biotinylated product. As a result, another tag is created on the recombinant protein which facilitates its efficient purification from cell lysates. Other tags such as FLAG or HA tags can be additionally introduced into the protein of interest. Thus, this approach in which single BirA and AP-Y are expressed in cell is mainly a modification of standard methods, such as immunopurification-mass spectrometry (IP-MS) or Tandem Affinity Purification and Mass Spectrometry (TAP-MS) with their inherent advantages and disadvantages. In the process of isolating and purifying protein complexes in the TAP-MS method from cell lysates, some partner proteins may be lost. This can be particularly common for weak protein–protein interactions, even in cases where buffer solutions used in washing steps of protocol are similar to those of physiological conditions. In contrary, the abundant proteins can also bind nonspecifically to the protein complex to be purified. All this leads to a large number of false-positive and false-negative results. For example, the use of IP-MS in Nanog partner protein analysis experiments allowed the identification of Oct4, but another important factor Sox2 was not detected [31].

Despite a slight change in design, the basic idea behind the second approach is that biotinylation itself is a direct result of the interaction or proximity of BirA-X and BAP-Y proteins in the cell. The PUB method uses a different acceptor peptide with less substrate specificity than Avitag to reduce background biotinylation and increase the difference from the control. This allows quantitative comparison of interactions, e.g., weakly (or transiently) interacting BirA-X and BAP-Y proteins and non-interacting BirA-X and BAP-Z proteins. Thus, the result of protein–protein interactions is the creation of a permanent covalent biotin mark on the partner protein in vivo, which overcomes the limitations of IP-MS and TAP-MS methods.

One practical application of in vivo biotinylation can be in the screening and selection of candidate substances that are regulators of protein–protein interactions. For example, it is possible to generate cell lines stably expressing BirA-X and BAP-Y in 96-well plates in which various low molecular weight compounds to be tested are added. The level of biotinylation, can be determined using a simple dot-blot method using streptavidin conjugates and select the most promising substances affecting protein–protein interactions.

Currently, a huge problem is the emergence of microbial strains resistant to a variety of antibiotics, such as tuberculosis pathogens. Protein–protein interactions also play a very important role in the physiology of *Mycobacterium tuberculosis* and the disease progress in the host [47]. Substances affecting these interactions could be the basis for a new generation of drugs.

BirA mutant versions, such as BioID [48] or TurboID [49], are now widely used to identify partner proteins in cells. The PUB method using wild type BirA can be complementary to these methods and provide additional information on cell organization and function.

## 5. Conclusions

Here we describe in detail the design and construction of BAP1070 and novel BAP1108 peptides. BAP1070 was used as a target for in vivo site-specific biotinylation by wild type BirA-fused proteins in the PUB method. A total of over 100 plasmid constructs, containing genes of transcription factors, histones, gene repressors, and other nuclear proteins were obtained during this work in projects related to this method [18,38,40,41,50]. The addition of a His-tag also allowed us to purify and detect all target peptides regardless of their biotinylation status, to perform normalization, and to refine biotinylation levels.

We expect that this method will also be applied to many cell types to uncover the roles of PTMs and to detail protein localizations in a spatiotemporal manner. The applicability of the PUB in living cells for quantification of PPIs will open the possibility of performing similar experiments in a tissue-specific manner in living animals, which can be used for testing new regulators of protein interactions.

## Figures and Tables

**Figure 1 life-12-00300-f001:**
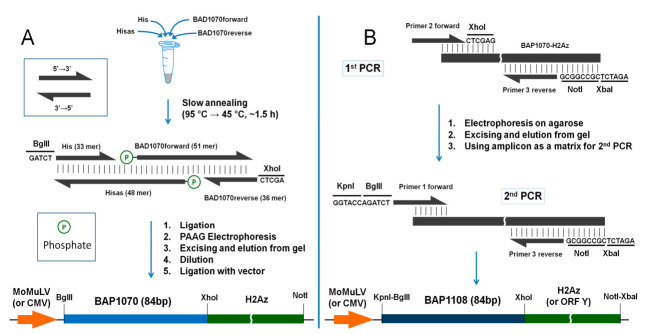
Workplans (**A**,**B**) for the generation of BAP1070 and BAP1108 fragments and subcloning into vectors for use as site-specific biotinylation targets in the PUB method. Note that His and BAD1070reverse primers in Workplan A already have cohesive BglII and XhoI ends.

**Figure 2 life-12-00300-f002:**
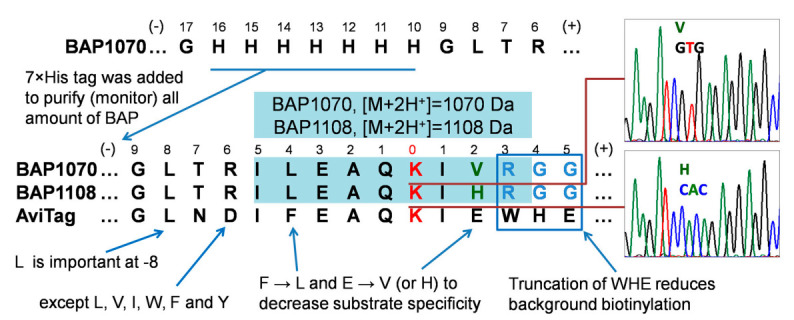
Alignment of sequences of BAP1070 and BAP1108 vs. Avitag and “consensus sequences” for biotinylation obtained by screening of combinatorial libraries [35,36]. Theoretical nonlabelled protonated tryptic peptides are highlighted by light blue with corresponding *m/z* values. The red K is the target lysine residue.

**Figure 3 life-12-00300-f003:**
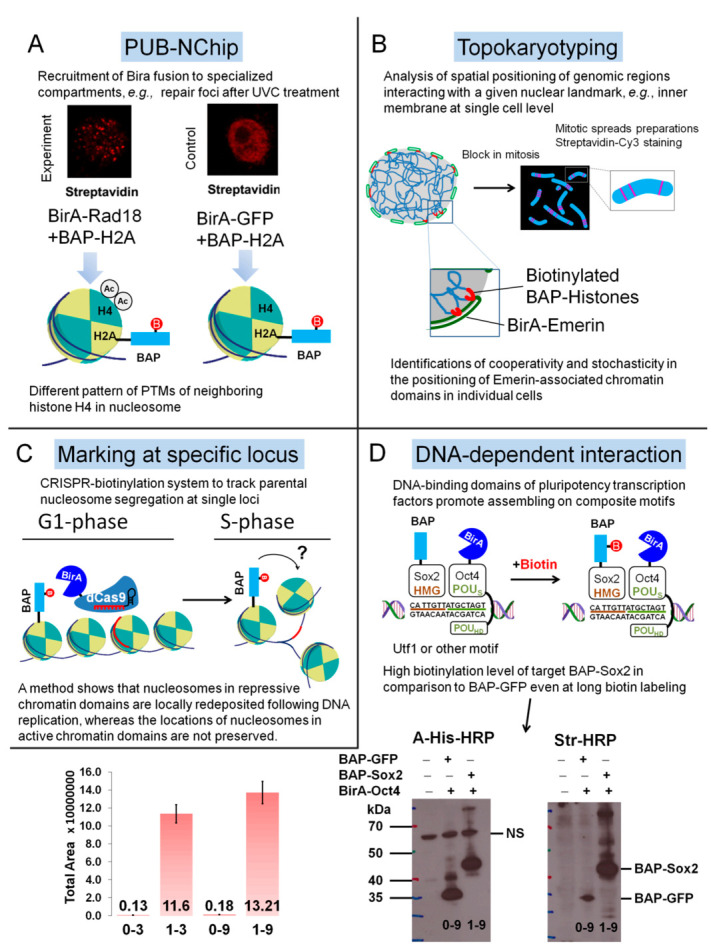
Different applications of BAP1070 peptide in site-specific biotinylation. (**A**) In PUB-NChip experiments BirA-Rad18 mainly found in DNA damage foci where it is recruited for monoubiquitination of PCNA protein, participating in DNA repair with other proteins. Different patterns of acetylation of H4 histone in nucleosomes, isolated from these foci in comparison with nucleosomes originated from other parts of chromatin were found. (**B**) The analysis of the spatial positioning of chromatin parts interacting with nuclear membrane protein BirA-Emerin for topokaryotyping. (**C**) Using the CRISPR-biotinylation system within specific loci allowed to track parental nucleosome and revealed a difference in the preservation of nucleosome deposition for repressed and active chromatin. (**D**) Biotinylation levels in PUB experiment with pluripotency transcription factors assembled on DNA motifs determined by analyzing MRM with Skyline and by Western blotting with a-His-HRP and Streptavidin-HRP. 0—BAP-GFP+BirA-Oct4, control. 1—BAP-Sox2+BirA-Oct4, experiment. Biotin labeling times 3 h (samples 0–3 and 1–3) and 9 h (samples 0–9 and 1–9). The average ratios 1–3/0–3 = 86 ± 6 and 1–9/0–9 = 71 ± 5 for three experiments.

**Table 1 life-12-00300-t001:** Primer sequences used for generation of biotin acceptor peptides BAP1070 and BAP1108.

Abbreviation	Primer Sequence (5′-3′ End)	Workplan
His	GATCTTGAACCATGGGACACCATCACCATCACC	A
Hisas	PhosATTCTTGTCAGGCCATGATGGTGATGGTGATGGTGTCCCATGGTTCAA	A
BAD1070 forward	PhosATCATGGCCTGACAAGAATCCTGGAAGCTCAGAAGATCGTGAGAGGAGGCC	A
BAD1070 reverse	TCGAGGCCTCCTCTCACGATCTTCTGAGCTTCCAGGA	A
Primer BADPCR	CATCATGGCCTGACAAGAATCCTG	A
Primer 1 NKpnIBAD	CACACACAGGTACCAGATCTTGAACCATGGGACACCATCACCATCACCATCATGGCCTGACA	B
Primer 2 NXhoIBAD	CATCACCATCATGGCCTGACAAGAATCCTGGAAGCTCAGAAGATCCACAGAGGAGGCCTCGAG	B
Primer 3CH2AZNotXbaIAS	TGTGTGTGTCTAGAGCGGCCGCTAGACAGTCTTCTGTTGTC	B

## Data Availability

The vector plasmids pcDNA3-BAP1070-Sox2 and pOz-BirA-GFP were deposited in Addgene repository (ID 133281 and 133283). The MS proteomics data have been deposited into the ProteomeXchange Consortium via the PRIDE partner repository, with the data set identifier PXD015756. Synthetic construct enhanced green fluorescent protein (eGFP) gene, partial cds (GenBank: MN443913.1).

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
