# Peer review of "Generation of Peptides for Highly Efficient Proximity Utilizing Site-Specific Biotinylation in Cells"

_life, 2022, doi:10.3390/life12020300_

Round 1
Reviewer 1 Report
In proposed study authors present application of tagged peptides in protein-protein interaction (PPI) study. Article is well written and highly informative. I recommend to accept the article after minor revision.
- Binding affinity between the peptides should be included. It will improve the importance of the current study.
- Correlation study should be performed with microarray, or LC-MS/MS to exhibit the importance of present study.
- Add following citations in current study to improve the article: https://doi.org/10.3390/pathogens8040173, https://doi.org/10.3390/biom10010142, https://doi.org/10.1038/nprot.2009.23
Author Response
Response to reviewers
Responses are displayed in blue
Reviewer: 1
Comments and Suggestions for Authors
In proposed study authors present application of tagged peptides in protein-protein interaction (PPI) study. Article is well written and highly informative. I recommend to accept the article after minor revision.
Reply: The authors thank very much the reviewer for his time to review the manuscript and his advises to improve it. I carefully followed these advises and I believe that the manuscript is now fairly improved.
- Binding affinity between the peptides should be included. It will improve the importance of the current study.
Reply: Data on the substrate specificity of BAP1070 and BAP1108 in comparison with Avitag were added to the text in section 4.3 and Supplementry Figure 1S.
- A correlation study should be performed with microarray, or LC-MS/MS to exhibit the importance of present study.
Reply: Description of LC-MS/MS experiments for BAP1108 was added to the section 4.3. Extracted ion chromatograms and MS/MS spectra were added to Supplementary Figure 2S.
- Add following citations in current study to improve the article: https://doi.org/10.3390/pathogens8040173, https://doi.org/10.3390/biom10010142, https://doi.org/10.1038/nprot.2009.23
Reply: These citations were added to the text in Discussion part and new section 4.5 of the manuscript.

Reviewer 2 Report
The authors present the experimental workplan with a good modularity for the generation of a new biotin acceptor peptide BAP1108 based on the widely applied original BAP1070. The discussion provides a well documented presentation of the different applications of BAP 1070 for site-specific biotinylation using the PUB method in the field of protein-protein interaction studies.
Since the new peptide BAP1108 differs by 3 aminoacids from the AviTag and by only one from the BAP1070, it would be interesting to have a comment on the expected performance of this new peptide.
line 138: 80-90 kb seems to be the incorrect size
Author Response
Reviewer: 2
Comments and Suggestions for Authors
The authors present the experimental workplan with a good modularity for the generation of a new biotin acceptor peptide BAP1108 based on the widely applied original BAP1070. The discussion provides a well documented presentation of the different applications of BAP 1070 for site-specific biotinylation using the PUB method in the field of protein-protein interaction studies.
Reply: The authors thank very much the reviewer for his time to review the manuscript and his advises to improve it. I carefully followed these advises and I believe that the manuscript is now fairly improved.
- Since the new peptide BAP1108 differs by 3 aminoacids from the AviTag and by only one from the BAP1070, it would be interesting to have a comment on the expected performance of this new peptide.
Reply: Discussion of properties of the new peptide BAP1108 were added to the text in the section 4.3 and Supplementry Figures 1S and 2S.
- line 138: 80-90 kb seems to be the incorrect size
Reply: Polyacrylamide gel electrophoresis was used to separate products resulted from annealing and ligation of four smaller oligonucleotides . We cut out the band with 84 bp, corresponding to BAP1070 fragment.
